# Systematic review of the healthcare cost of bronchopulmonary dysplasia

Jhangir Humayun [ID] ,[1,2] Chatarina Löfqvist,[1,2,3] David Ley,[4] Ann Hellström [ID] ,[3] Hanna Gyllensten [ID] [1,2]

[1]Institute of Health and Care Sciences, University of Gothenburg, Gothenburg, Sweden
[2]Centre for Person-Centred Care - GPCC, University of Gothenburg, Gothenburg, Sweden
[3]Department of Clinical Neuroscience, Institute of Neuroscience and Physiology, University of Gothenburg, Gothenburg, Sweden
[4]Department of Pediatrics, Institute of Clinical Sciences Lund, Lund University and Skåne University Hospital, Lund, Sweden

**Correspondence to**
Professor Hanna Gyllensten;
hanna.gyllensten@gu.se

## ABSTRACT

**Objectives** To determine the costs directly or indirectly related to bronchopulmonary dysplasia (BPD) in preterm infants. The secondary objective was to stratify the costs based on gestational age and/or birth weight.

**Design** Systematic literature review.

**Setting** PubMed and Scopus were searched on 3 February 2020. Studies were selected based on eligibility criteria by two independent reviewers. Included studies were further searched to identify eligible references and citations.

Two independent reviewers extracted data with a prespecified data extraction sheet, including items from a published checklist for quality assessment. The costs in the included studies are reported descriptively.

**Primary outcome measure** Costs of BPD.

**Results** The 13 included studies reported the total costs or marginal costs of BPD. Most studies reported costs during birth hospitalisation (cost range: Int$21 392–Int$1 094 509 per child, equivalent to €19 103–€977 397, in 2019) and/or during the first year of life. One study reported costs during the first 2 years; two other studies reported costs later, during the preschool period and one study included a long-term follow-up. The highest mean costs were associated with infants born at extremely low gestational ages. The quality assessment indicated a low risk of bias in the reported findings of included studies.

**Conclusions** This study was the first systematic review of costs associated with BPD. We confirmed previous reports of high costs and described the long-term follow-up necessary for preterm infants with BPD, particularly infants of very low gestational age. Moreover, we identified a need for studies that estimate costs outside hospitals and after the first year of life.

**PROSPERO registration number** CRD42020173234.

## INTRODUCTION

Bronchopulmonary dysplasia (BPD) is a severe lung condition that can affect new borns. It is prevalent in premature infants, particularly infants that require oxygen therapy. Infants born more than 10 weeks preterm are more susceptible to BPD because they often require a long treatment period with different types of breathing support, such as nasal continuous positive airway pressure, ventilators, different bronchodilators and steroids.[1]

### Strengths and limitations of this study

► This is the first systematic review specifically focused on costs resulting from bronchopulmonary dysplasia after preterm birth.
► Authors searched systematically through PubMed and Scopus, which are appropriate databases for the investigation topic, and also reviewed reference lists and citations of the relevant papers to identify any additional studies.
► The main limitation was the lack of search for unpublished and/or grey literature.

The definition of BPD has varied considerably over time. The condition was first described by Northway et al, as a radiographic pattern of lung injury observed in prematurely born infants after the use of extensive mechanical ventilation.[2] Considerable strides in care have been made for these infants over the past decades, including, but not limited to, the use of antenatal steroids, surfactant therapy, vitamin A supplementation and caffeine treatment.[3–5] The administration of antenatal steroids and exogenous surfactant has dramatically reduced the severity of respiratory distress syndrome, and consequently, increased the survival of preterm infants.[6] Before the widespread use of antenatal steroids and surfactant, BPD in preterm infants was mainly attributed to lung injury caused by barotrauma and oxygen toxicity.[6] The pathogenesis of the BPD observed currently, in the post surfactant era, is multifactorial; various causes have been found, including extreme prematurity, infection, nutritional deprivation and the need for prolonged mechanical ventilation.[7 8] Short-term management thus includes, preferably non-invasive, respiratory support titrated to specific oxygen saturation ranges, exogenous surfactant therapy, respiratory stimulants such as caffeine, corticosteroids to reduce inflammation and inhaled nitric oxide against pulmonary hypertension (PH). In contrast, long-term management includes a

larger range of products to handle persistent and severe chronic respiratory disease and related comorbidities.[9] It is imperative to understand this new form of BPD and its effects during infancy and beyond because the survival of preterm infants is increasing.[10]

Currently, BPD is commonly defined as a condition that requires oxygen administration beyond 36 weeks postmenstrual age (PMA). BPD severity is classified as light, moderate or severe depending on the required respiratory support and inhaled oxygen concentration. Comorbidities and maternal infections increase the risk of developing severe BPD after preterm birth. Disease severity varies considerably; for example, in mild cases, infants might only present with an accelerated respiratory rate; but mechanical ventilation may be required for a prolonged period in severe cases.[1] Indeed, serious conditions often require continuous oxygen therapy for several months.

The incidence of BPD seems to be inversely related to the gestational age (GA, reported in weeks) at birth. Studies that included infants born at low GAs have reported higher incidences of BPD.[11] The National Institute for Child Health and Human Development neonatal network periodically reports the incidence of BPD. They define BPD as a condition that requires oxygen administration at 36 weeks PMA. They reported BPD incidences of 52% in infants with birth weights (BW) of 501–750 g, 34% in infants with BW of 751–1000 g and 15% in infants with BW of 1001–1200 g. Among infants that weighed 1201–1500 g, the incidence was as low as 7%.[12]

Fetal growth restriction is commonly defined as low BW, corrected for GA. Fetal growth restriction has a clear impact on the incidence and severity of BPD. Studies have shown that among infants born at a similar GA, infants that were small for their GA (SGA) had a higher BPD incidence than infants that had normal BW for their GA.[11] Globally, the survival rates of extremely preterm infants (GA <28 weeks) have varied considerably, according to the economic setting. Highly developed countries have reported extremely preterm infant survival rates as high as ~90%, but it can be as low as ~10% in low-income settings.[13] Similarly, the incidence of BPD in preterm infants varies considerably with the degree of economic development in the country of birth.

The effects of BPD on infant health extend beyond the initial hospitalisation. Infants with BPD are hospitalised to a much greater extent than infants without BPD.[14] Lung function abnormalities, such as small airways, often persist for an extended period. Long-term effects, such as reduced lung function and obstructive airway disease, are more prevalent among individuals born prematurely than their born full-term peers. Among preterm infants, those with BPD exhibit a greater need for respiratory medications than those without BPD.[14] Neurodevelopment can be severely affected by BPD. Infants with BPD have an increased rate of abnormal neurodevelopment, including cognitive impairments and cerebral palsy. In severe cases of BPD, movement morbidities can occur, like chorea and akathisia.[15]

An economic evaluation is essential for aiding decision-making bodies in prioritising healthcare funding effectively and understanding the associated long-term costs. A review[15] was previously published on the healthcare and societal cost of BPD, but the search was not systematic; the authors handpicked all the articles. To our knowledge, no systematic review has been published that discussed the healthcare costs of BPD. Thus, there was a need to compile and present the available data on this subject in a systematic review.

The present review aimed to present all available data that pertained to the costs related to BPD, directly or indirectly, in preterm infants. The secondary objective was to stratify those costs based on the GA or BW of the infant.

## METHODS
This study was conducted according to the Preferred Reporting Items for Systematic Reviews and Meta-Analyses recommendations.[5]

### Search strategy
In collaboration with personnel of the Medicine Library at the University of Gothenburg, the authors determined that the PubMed and Scopus databases were the best choice for this review. The search was conducted on 3 Feb 2020 in both databases. Search terms (online supplemental table S1) were drafted and discussed individually for both databases. The search was performed, and articles were included according to the criteria listed below. No restrictions were applied for publication date, country of origin or language. Included studies were further searched to identify eligible references and citations.

### Study selection
We included observational and interventional studies that investigated the costs associated with BPD. The study population comprised preterm human infants with BPD. Two reviewers (JH and HG) independently screened the titles and abstracts of identified studies to assess their eligibility for inclusion. After reviewing the titles and abstracts, duplicates were removed. Then, the reviewers obtained full texts of potentially eligible articles and assessed them independently to determine which articles met the predetermined inclusion criteria. When the reviewers were uncertain of the eligibility of a study, a third reviewer (CL) participated in the decision.

### Data extraction and quality assessment
To ensure essential data were extracted consistently from relevant articles, we created an extraction table. Data from eligible studies were independently extracted by two reviewers (JH and HG). Any differences were resolved by discussion and consultation with a third reviewer (CL). The predesigned data extraction table included but was not limited to the following

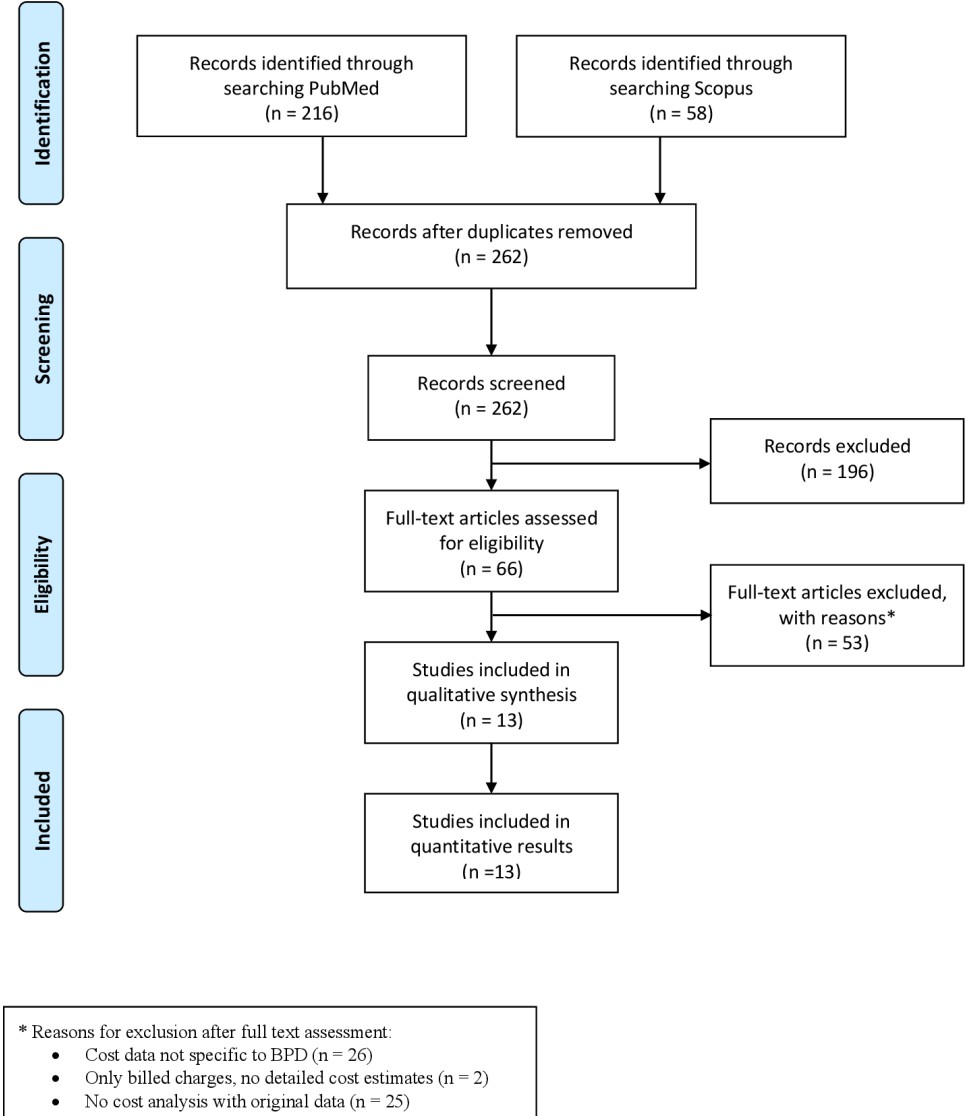

**Figure 1** Flow diagram shows the study selection process, following the PRISMA guidelines. BPD, bronchopulmonary dysplasia; PRISMA, Preferred Reporting Items for Systematic Reviews and Meta-Analyses.

data: the name of the first author, year of publication, study design, location, sample size, healthcare costs and description of the costs included. When available, healthcare costs were stratified according to the GA at birth or BW. The two authors independently assessed the quality of the selected studies, based on predetermined quality assessment criteria for health economic studies (the Consensus on Health Economic Criteria, CHEC-list) described previously by Evers *et al.*[16] Although designed for economic evaluations, and this CHEC-list has previously been used in similar studies.[17 18]

### Analyses
Costs were converted to USD, based on the purchasing power parities of each year.[19] Then, the costs were inflated to 2019 values with the Organisation for Economic Co-operation and Development inflation factor.[20] Costs were analysed, based on the chronological

age of the participants included in each study and by their GA or BW.

### Patient and public involvement
This project did not include patient or public involvement in developing the research questions, design, conduct, choice of outcome measures or recruitment.

### RESULTS
The initial search results yielded 274 articles. After removing duplicates (figure 1), 262 unique articles were screened, and 195 were excluded based on the abstract or title. Among the 66 full articles screened, 13[21–33] unique studies met the inclusion criteria. The hand search of references and citations of included studies did not identify any additional eligible studies. The included studies were conducted in 1981–2015 (table 1), in the USA (seven

**Table 1** Overview of studies included in this review

| # | First author (year) | Country (study period) setting | Study design | BPD definition | Sample size | Time horizon: age of children | Healthcare cost per child with BPD | Cost perspective: cost inclusion |
|---|---|---|---|---|---|---|---|---|
| 1 | Lapcharoensap (2020)[33] | USA (2008–2011) California Perinatal Quality Care Collaborative | Retrospective cohort | Oxygen requirement >36 weeks PMA | BPD: 2696 Controls: 5302 | 1 year: from birth, including index hospitalisation | US$442 468 | Hospital: Direct hospital costs calculated from charges in discharge records using cost-to-charge ratio |
| 2 | Mowitz (2019)[32] | USA (2009–2015) Premier Perspective Database | Retrospective cohort | ICD code | BPD: 4904 Controls: 7113 | Short term: Index hospitalisation | US$225 204 | Hospital: Direct hospital costs, from billed services |
| 3 | Álvarez-Fuente (2017)[29] | Spain (2013) SEN1500 database | Retrospective cohort: Model base case for minimum cost of BPD using weighted costs | Oxygen dependence >28 days after birth (also >36 weeks for severity assessment) | BPD: 2628 | 2 years: from birth, including index hospitalisation | €45 049–€118 760 | Health system: Hospital costs and follow-up cost; inpatient costs were estimated using diagnosis related group weights, price lists were used for outpatient visits and immunisation |
| 4 | Patel (2017)[30] | USA (2009–2012) Rush University Medical Centre NICU, Chicago/ Illinois | Prospective cohort (compare own mothers milk v. formula) | Oxygen requirement >36 weeks PMA | BPD: 77 Controls: 177 | Short term: Index hospitalisation | US$269 004 (median) | Hospital: Hospital costs, overhead and physician payments from a financial data repository with costs for chargeable items |
| 5 | Stevens (2017)[31] | USA (2008–2011) The California Office of Statewide Health Planning and Development (OSHPD) database | Retrospective cohort: Regression model to identify the cost incurred by infants with specific conditions | ICD code | Total: 2 021 013 BPD: 2189 | 1 year: from birth, including index hospitalisation | US$133 140 Cost increase=22.4% | Hospital: Commercial claims data from the region corrected to national costs using register data from the Healthcare Cost and Utilisation Project data set |
| 6 | Johnson (2013)[28] | US (2005–2009) Rush University Medical Centre NICU, Chicago/ Illinois | Retrospective cohort | Oxygen dependence >28 days after birth (also >36 weeks for severity assessment) | BPD: 230 Controls: 195 | Short term: Index hospitalisation | US$103 151 | Hospital: Hospital costs excluding physician fees from a financial data repository with costs for chargeable items |
| 7 | Landry (2012)[27] | Canada (1983–2008) MED-ECHO and RAMQ databases, Quebec | Retrospective cohort | ICD code | BPD: 773 Controls: 2669 | 16–25 years: from birth, after the index hospitalisation | $C13 472 | Health system: Hospitalisation follow-up cost and prescription drugs; inpatient costs were estimated using a case mix approach, billed fees were used for outpatient visits and drugs |
| 8 | Greenough (2011)[26] | UK (2002–2005) Previously presented cohort, hospitals not stated[49] | Retrospective cohort | Oxygen dependence >28 days after birth | BPD: 160 | 2 years: 5–7 years of age | £523–£705 (median) | Health system: Hospital admission costs*, follow-up cost, and prescription drugs; price lists were used for admissions and drugs, unit costs based on wages and time used by clinicians for outpatient visits |

**Table 1** Continued

| # | First author (year) | Country (study period) setting | Study design | BPD definition | Sample size | Time horizon: age of children | Healthcare cost per child with BPD | Cost perspective: cost inclusion |
|---|---|---|---|---|---|---|---|---|
| 9 | Greenough (2006)[25] | UK (1999–2002) Previously presented cohort, hospitals not stated[49] | Retrospective cohort | Oxygen dependence >28 days after birth | BPD: 190 | 2 years: 2–4 years of age | £950–£3054 (median) | Health system: Hospital admission costs*, follow-up cost, and prescription drugs; price lists were used for admissions and drugs, unit costs based on wages and time used by clinicians for outpatient visits |
| 10 | Akman (2002)[24] | Turkey (1999–2000) Marmara university hospital, Istanbul | Retrospective cohort | Oxygen requirement >36 weeks PMA | BPD: 72 | Short term: Index hospitalisation | US$14 810 (marginal cost) | Hospital: Hospital charges collected from the hospital Accounting Department, including physician and nursing services, laboratory and radiology tests and medical supplies |
| 11 | Miller (1998)[23] | USA (1990) Tulane University Medical Centre, Louisiana | Retrospective cohort | Oxygen dependence >28 days after birth and chest X-ray | BPD:89 | 4.5 years: from birth, after the index hospitalisation | US$19 034† | Payer (third party and parents): Direct health costs (registered charges, or costs translated to charges using cost-to-charge ratios), non-health costs (expenses), and indirect cost (lost income) to families |
| 12 | McAleese (1993)[22] | USA (1981–1989) Dartmouth-Hitchcock Medical Centre, New Hampshire | Retrospective cohort | Oxygen dependence >28 days after birth and chest X-ray | BPD: 59 | 2 months to 2.7 years: from birth, including index hospitalisation | Initial hospitalisation = US$197 668 Home care = US$4262 to US$68 136 (medians, by group) Indirect costs = US$650–US$132303 (range) | Payer (third party and parents): Hospital charges for hospitalisation, charges for follow-up cost and home care equipment. expenses and travel costs, and indirect cost (lost income) to families |
| 13 | Monset-Couchard (1984)[21] | France (1981) Port-Royal Intensive Care Unit, Paris | Retrospective cohort | Hospital data | Not stated | Short term: Index hospitalisation | ₣336 545 | Hospital: Unit costs for hospitalisations, travel charges (including ambulance and air travel) |

Unless otherwise stated, the costs are the mean costs reported in the original paper.
*Does not include index hospitalisation since the year of birth is not included in the time period of each study.
†Mean cost was computed based on the data available in the article.
BPD, bronchopulmonary dysplasia; ICD, International Classifications of Diseases; PMA, postmenstrual age.

studies),[22 23 28 30–33] UK (two studies),[25 26] Spain,[29] Canada,[27] Turkey[24] and France.[21]

Six studies[22 23 25 26 28 29] defined BPD as oxygen dependence beyond 28 days after birth (table 1), and two of these studies[28 29] also classified BPD as mild/moderate/severe, based on whether oxygen was administered at 36 weeks PMA, which, according to Jobe and Bancalari,[34] indicated moderate/severe BPD. Three studies[24 30 33] used only the 36 weeks PMA oxygen use definition, while three additional studies[27 31 32] defined BPD based on the International Classifications of Diseases 9 (ICD.9) (770.7) code from a registry, without reference to GA or time since birth at diagnosis. The oldest included study,[21] was based on hospital data, not stating whether they used ICD codes or what time period they used for the oxygen requirement. However, their definition was consistent with the classical definition of BPD,[2] which was based on clinical signs. In two studies,[22 23] the definition of BPD required a chest X-ray consistent with BPD, in addition to the oxygen dependence beyond 28 days after birth requirement.

### Risk of bias in included studies

Potential sources of bias included the lack of participant characteristics in some of the included studies. Additionally, four studies[21–24] lacked information about potential conflicts of interest for the participating researchers (online supplemental table S2). Only one study[31] conducted a sensitivity analysis of their results. None of the included papers clearly stated the research question, but it could be gleaned from the objective of each study. Three studies[25 26 30] included comparisons between groups or long-term outcomes, and no study included an evaluation of health outcomes. Thus, some of the CHEClist items were not suitable for evaluating these studies. Checklist items judged relevant were largely covered by the included papers. Only three papers[21 23 24] had more than three items missing, and the items not covered throughout the papers mainly were related to lack of a well-defined research question and lack of sensitivity analyses. Overall, the assessment suggests a low risk of bias in the included papers.

### Total costs for BPD

A majority of studies[21 22 24 28–30 32 33] (table 2) reported costs during the birth hospitalisation and/or the first year of life (eight studies). One study[29] reported follow-up costs during the first 2 years of life, including birth hospitalisation. Two other studies[25 26] reported 2 years follow-up costs for age groups 2–4 and 5–7 years, respectively. Two studies[23 27] included a long-term follow-up, with an average follow-up of 4.5 years and 19.3 years, respectively (figure 2).

### Index hospitalisation

The mean costs during the index hospitalisation ranged from Int\$21 392 to Int\$1 094 509 (figure 3). For five studies[22 28 30 32 33] conducted in the USA, the range of

index hospitalisation costs was Int\$1 22 983–Int\$466 216. If only including those four US studies limiting inclusion to severe BPD, the range is Int\$242 995–Int\$466 216.[22 30 32 33]

### Hospital readmissions in the first two years of life

One study[33] analysed rehospitalisations among infants with BPD. They showed that rehospitalisations among infants with BPD occurred more than twice at frequently and incurred higher costs than rehospitalisations among their peers without BPD. Rehospitalisations were required in 25.1% of infants with BPD, and the mean cost was estimated to be Int\$46 948 during the first year of life.[33] One study[29] estimated that the cost associated with BPD for a mean follow-up of the first 2 years of life was Int\$14 084.

### Follow-up costs and home care

Two studies[25 26] reported costs during a specific age, thus not including the costs occurring at birth and during the first years of life: Among children with BPD, the median costs during 2 years of follow-up for ages 2–4 were estimated at Int\$5556 for those that required home oxygen after the initial discharge, compared with Int\$1728 for those that did not require home oxygen after the initial discharge from the hospital.[25] Furthermore, the median costs during 2 years of follow-up for ages 5–7 were Int\$1135 for those that required home oxygen, compared with Int\$842 for those that did not require home oxygen after the initial discharge.[26] Children initially requiring oxygen during a longer period also required more resources at older ages.

Other studies reported costs from birth and onwards, but excluding the index hospitalisation: One study[23] reported that the median follow-up cost of healthcare for children with BPD 4.5 years after initial hospitalisation was Int\$25 268 for inpatient care and Int\$6625 for outpatient services. One study[27] analysed the cost of healthcare for children with BPD during a follow-up for ages 16–25 years; they reported that the annual cost was Int\$12 973 per person.

Moreover, one study[22] reported that the median cost of home healthcare (including indirect costs for lost wages) was Int\$133 427 with a nurse and Int\$8346 without a nurse. Thus, costs for home healthcare were largely dependent on the possibilities to employ a nurse based on private insurance payments and other funds.

### Indirect costs

One study[23] reported an indirect cost of Int\$2800 during 4.5 years after the initial hospitalisation. One study[22] reported indirect costs as part of the home care costs, and for the 30 families reporting the loss of wages, this corresponded to a range of Int\$516–Int\$44 105 per family (with 9–577 days of lost income) in 2019 value.

### Increase in costs due to BPD

In four of the included studies,[24 28 32 33] BPD incurred, on average, 31%–241% extra expenses in healthcare. One study[27] from Canada reported that the mean follow-up

cost increased by Int$2651/year, a 26% increase over the cost for preterm peers without BPD.

In four studies[28 29 32 33] that reported costs according to GA categories, the cost ranges were: Int$196713–Int$507638 for those born at or before GA 24 weeks/<750 g; Int$149071–Int$453633 for those born at GA 25–26 weeks/750–999 g; and Int$73237–Int$351027 for those born at GA 27–28 weeks/1000–1249 g. One study[33] reported a cost of Int$313223 for those born at GA 29 weeks or later/>1249 g (figure 3).

**Table 2** Costs for infants with BPD

| Reference subgroups reported in the paper | Total cost (Int$) | Cost of index Hospitalisation (Int$) | Cost of other care (Int$) | Cost increase over controls due to BPD, Int$ (%) |
|---|---|---|---|---|
| 1. Lapcharoensap (2020)[33] | 477901 | 466216 | Rehospitalisation: 46948 | 246284 (106%) |
| ≤24 weeks | 507638 | | | 118808 (35%) |
| 25–26 weeks | 453633 | | | 129609 (40%) |
| 27–28 weeks | 351026 | | | 124209 (55%) |
| >28 weeks | 313223 | | | 151211 (93%) |
| 2. Mowitz (2019)[32] | See index hospitalisation | 242995 | | 58381 (31%) |
| ≤24 weeks | See index hospitalisation | 326962 | | 26975 (9%) |
| 25–26 weeks | See index hospitalisation | 247901 | | 32370 (15%) |
| 27–28 weeks | See index hospitalisation | 190983 | | 36686 (24%) |
| >28 weeks | – | – | | – |
| 3. Álvarez-Fuente (2017)[29] | 74618–196713 | | 2 years follow-up: | – |
| <750 g | 196713 | 182627 | 14084 | – |
| 750–999 g | 149071 | 134986 | 14084 | – |
| 1000–1249 g | 73237 | 60534 | 14084 | – |
| >1249 g | – | – | – | – |
| 4. Patel (2017)[30] | See index hospitalisation | 290546* Marginal cost: 45287* | | 164092* |
| 5. Stevens (2017)[31] | 148251 | – | – | (22.4%) |
| 6. Johnson (2013)[28] | See index hospitalisation | 122983 | | 69969 (132%) |
| <750 g | | | | 51639 |
| 750–999 g | | | | 41196 |
| 1000–1249 g | | | | 29317 |
| >1249 g | | | | 23810 |
| 7. Landry (2012)[27] | 12973/year | | 16–25 years follow-up: 12973/year | 2651/year (26%) |
| 8. Greenough (2011) (2 years at 5–7 years of age)[26] | Home oxygen: 1135* No oxygen: 842* | | Home oxygen: 1,135* No oxygen: 842* | |
| 9. Greenough (2006) (2 years at 2–4 years of age)[25] | Home oxygen: 5556* No oxygen: 1728* | | Home oxygen: 5556* No oxygen: 1728* | |
| 10. Akman (2002)[24] | See index hospitalisation | 21392 | – | 15116 (241%) |
| 11. Miller (1998)[23] | 34692 | | 4.5 year follow-up Inpatient services: 25268 Outpatient services: 6625 Indirect costs: 2800 | |
| 12. McAleese (1993)[22] | See index hospitalisation | 387085 | Home care, nurse: 133427* Home care, no nurse: 8346* (including indirect costs) | |
| 13. Monset-Couchard (1984)[21] | See index hospitalisation | 1094509 | | |

Values are the mean costs unless otherwise stated.
Int$=The cost after converting to purchasing power parities, then inflating to the 2019 value.
*Median cost.
BPD, bronchopulmonary dysplasia.

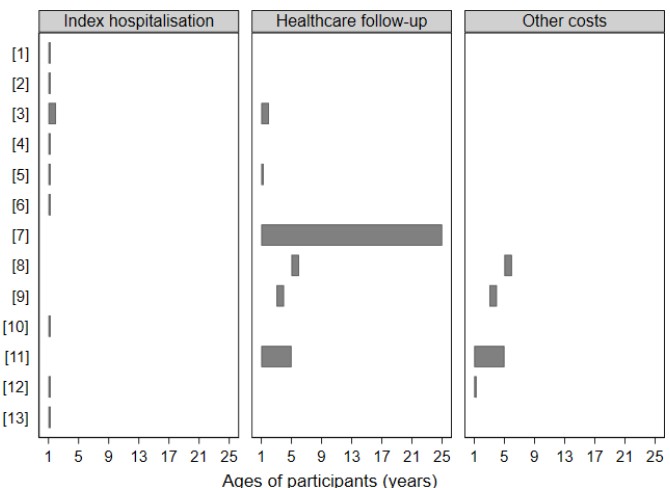

**Figure 2** Overview of the costs of BPD and the study period included in each article. The numbers on the Y-axis refer to the numbers assigned to each study (table 1).

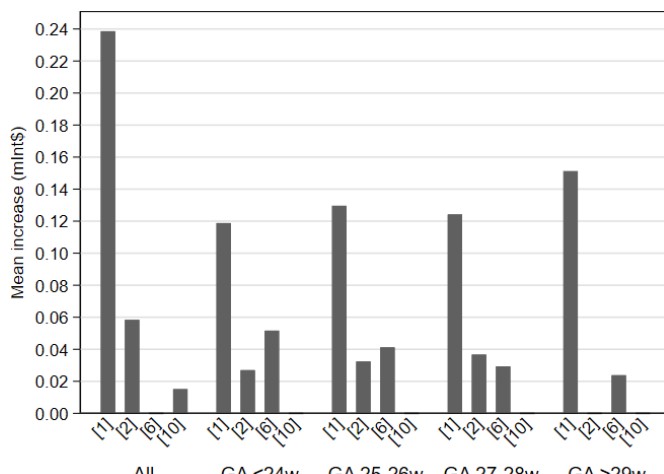

**Figure 4** Mean cost increase for infants with BPD, stratified by gestational age (GA) at birth. The numbers on the x-axis refer to the numbers assigned to studies in table 1. BPD, bronchopulmonary dysplasia.

Higher cost increases were reported for infants with BPD born at higher GAs than those born at lower GAs. For example, studies reported that the mean costs for infants with BPD increased by 9%–35% for those born at or before GA 24 weeks; by 15%–35% for those born at GA 25–26 weeks; and by 19%–35% for those born at GA 27–28 weeks. One study[33] reported a 93% cost increase for infants born at GA 29 weeks or later. The mean total cost increase for infants with BPD, stratified by GA, is shown in figure 4.

## DISCUSSION

The included studies reported high costs for preterm infants with BPD during the initial hospitalisation after birth (range: Int$21 392–Int$1 094 509 per child, equivalent to €19 103–€977 397, in 2019). However, looking at only four studies conducted in one country and similar

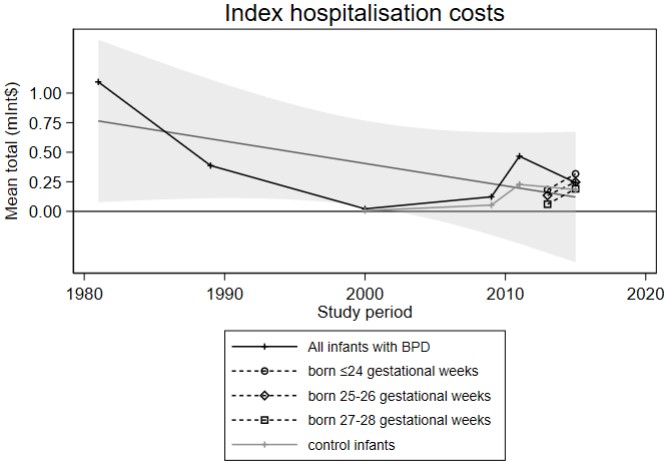

**Figure 3** Costs of the index hospitalisations for preterm infants with BPD in studies conducted between 1981 and 2015. All costs were converted to Int$, based on purchasing power parities, then the values were inflated to 2019 values. abbreviations: BPD, bronchopulmonary dysplasia.

inclusion criteria, the range was Int$242 995–Int$466 216. Moreover, the studies reported high use of resources associated with BPD during the following years, up to school age. The highest mean costs were associated with infants born at extremely low GAs.

Several prior reviews and modelling studies in this field[15 35] have reported costs, but they based study inclusions on expert opinions and targeted searches. Therefore, the present study provided an overview of all the evidence available on costs for BPD, with expected low risk of bias in included studies based on the quality assessment tool. However, the main limitation of this study was the lack of searches for identifying grey literature and not performing analysis of publication bias. However, we did perform an initial search based on references identified in, for example, modelling studies and reviews, when we designed the search strategy to identify all the relevant publications with our search strategy. We found several studies that addressed costs of interventions, such as immunisation or infection treatment for infants with BPD, but in those cases, the costs were not included for BPD per se. Our search was explicitly tailored to identify articles on BPD; thus, we excluded articles that used an alternate term, like chronic lung disease. We identified very relevant articles that were relevant, considering the great need for quality long-term healthcare for preterm infants with BPD.

Moreover, we initially intended to conduct a meta-analysis of results, but we found that, in the separate papers, the costs reported differed too extensively in content and time-horizons to support that type of analysis. For the same reason, we could not conduct a formalised synthesis of the results. Moreover, the recently published guidelines for a data synthesis without a meta-analysis[36] were unsuitable for this study; instead, they were more suited to the synthesis of effect estimates for intervention studies. Consequently, the analysis herein was descriptive.

Another factor making comparisons between results in included studies was that, in the BPD literature, the operational definition of BPD has varied since the condition was first discovered.[9] Thus, the extent of variability we encountered in the included papers (detailed in the Results section) might have affected the selection of infants, which might have influenced the healthcare costs. It is also important to note that the costs of BPD reported in the selected studies was based on the costs incurred by oxygen administration. However, the full economic impact of BPD should also include the costs associated with preventing BPD development in infants at risk. This same limitation applies to all retrospective studies that based the definition of BPD on ICD codes and clinical assessments because infants who successfully avoided BPD development would not be classified as BPD, either with the codes or in clinical assessments.

Regarding the cost estimations, the majority of the included articles focused on the initial hospitalisation. However, after the initial hospitalisation, many infants with BPD require rehospitalisation during the first years of life due to respiratory illnesses.[33] Infants born prematurely often display symptoms and signs of obstructive airway disease that persist for prolonged periods; thus, it is crucial to implement standardised, regular follow-up schedules.[37]

Our findings showed how the cost estimates for BPD varied across different countries and during different periods. Table 2 shows a large cost discrepancy among different countries, for example, Turkey and the USA, which several factors might explain. First, BPD treatment programmes might differ with different healthcare providers, during different years and in different countries.[38] For example, articles that investigated the costs of BPD in the USA reported much higher costs than those reported in other countries; this discrepancy might partly be explained by the higher cost of healthcare, in general, in the USA.[39] Among studies conducted in the USA, the study by Johnson et al[28] stands out compared with lower costs for the initial hospitalisation, possibly due to the exclusion of physician fees or from the BPD definition being inclusive also of milder cases. Second, the included studies spanned a long time period, where healthcare will have changed dramatically due to technological development. It is clear from the data provided that care of infants with BPD has been resource intense throughout the period, and the inflated estimates indicate the current value of the resources used at that time, but it is difficult to compare costs between a study from the early 1980s to those conducted in 2010s. The patient group will look very different, with more infants born very early surviving to develop BPD. In addition to the potential effects of different BPD definitions, infants with BPD frequently have other comorbidities, such as brain injury and retinopathy of prematurity (ROP), which can lead to functional impairments that impact the cost of healthcare.[40] The cost of BPD without any comorbidity can be difficult to differentiate from the total cost, including comorbidities. For example, comorbidities like PH in infants with BPD can significantly increase healthcare costs due to, more extended hospitalisations and a higher likelihood of requiring home respiratory support.[41] Third, healthcare quality can differ between different healthcare providers, resulting in cost differences. Fourth, some of the included articles omitted data for some costs, for example, the cost of physicians.[28] Finally, as previously discussed, an SGA condition can exacerbate the severity of BPD; this condition was not considered in the cost calculations because the included articles did not report the combined effects of GA and BW.

Two studies compared healthcare costs between preterm infants with BPD and their preterm peers without BPD. Those data enabled a calculation of the cost increase due to BPD. The results indicated that the cost differences were higher among infants born at lower GAs. This finding was consistent with cost estimates for other diagnoses in the same patient groups, such as infants born to mothers with preeclampsia.[31] Among our included studies, the study by Johnson et al,[28] estimated marginal costs in infants with BPD compared with infants of the same weight without any of four major comorbidities (sepsis, brain injury, necrotising enterocolitis (NEC) or BPD). They found that the cost increase was more prominent among the smallest infants (<750 g). Lapcharoensap et al,[33] compared infants with BPD to all other infants of the same age (no limitation on comorbidities). They reported the opposite relationship, where, more considerable cost differences were observed among those born at higher GAs. This relationship might be expected because, in the reference groups, infants born at higher GAs can be expected to have fewer of the major comorbidities than infants born at lower GAs; thus, the high GA group would show a larger cost difference between infants with and without BPD. This effect was also reported by Mowitz et al,[32] although the detailed cost data were not available. They reported that among all children (regardless of comorbidity), the cost increase due to BPD was largest for infants born at a higher GA; however, when infants with BPD were compared with those without BPD, ROP or intraventricular haemorrhage (IVH), the cost increase was similar independent of GA. In contrast, the study by Johnson et al showed a clear trend towards a higher cost difference due to BPD among the smallest infants. The differences between the two latter studies might be explained by the fact that the comorbidities excluded in the study by Johnson et al[28] (the comparison group had no sepsis, brain injury, NEC or BPD) were different from those excluded in the study by Mowitz et al[32] (the comparison group had no ROP, IVH or BPD).

It is clear from the included studies that infants and children with BPD often required long-term follow-ups. This requirement affects the lives of these infants in various direct and indirect ways, which have not been adequately investigated. A recent modelling study,[35] based on a targeted literature search, estimated the lifetime burden of BPD for infants born ≤28 weeks GA. That

study explored how BPD severity and major complications affected other aspects of life, like clinic visits for wheezing episodes, asthma exacerbations, psychiatric illnesses, etc. As expected, the model found that BPD had a large impact on healthcare use, quality-of-life and survival, but it also showed that the degree of impact was largely dependent on BPD severity. Accordingly, improvements in BPD therapy might substantially reduce the impact of BPD.[35] However, as those authors acknowledged, the model could only use the available data. Thus, it was limited by, for example, data on each complication separately (not accounting for concomitant diagnoses in one patient, potentially increasing costs more than expected from single diagnosis studies). That study also reported that the healthcare cost of BPD for the first year of life was $C197 100, which was consistent with the estimates of more recent studies included in this review.

The findings from both our systematic review and the previous modelling study[35] suggested a need for more research on the long-term economic impact of BPD. Preferably, those studies should also include costs outside the health system, such as costs due to changes in parent productivity and costs for schooling and daycare. In countries like the Nordics, where much information about resource use is available in national and/or regional registers collected for administrative purposes and available for research, many relevant resources can be studied in retrospective register studies.[42 43] However, some types of information are less likely to be included in registers, such as informal care provided by parents and other relatives. Thus, studies using prospective and retrospective data collection methods should add important knowledge on the long-term costs associated with BPD.

Moreover, studies should consider the costs related to the combined impact of complications. It has been argued that studies with a societal perspective can better identify potential inequities and inefficient use of society's resources.[44] One can speculate that one reason for the paucity of cost studies on BPD is that new drug developments have slowed after the success of surfactant therapy, which was introduced in the early 1990s.[45] Although several advances in the care of preterm infants and BPD have improved survival and symptom reduction,[46] these care applications and treatments have not, to the same extent, been subjected to economic and cost assessments.[47] To minimise the growing burden of BPD, it is vital to prevent prematurity in the first place, and in the second place, to develop new treatments for prematurity-related diseases.[48] The ongoing research on the administration of stem cells, anti-inflammatory agents and insulin-like growth factor I (IGF-I)/insulin-like growth factor-binding protein 3 (IGFBP-3) among extremely preterm infants[9] may spark a renewed interest in how resources are best used for this patient group.

This systematic review confirmed previous reports that BPD in preterm infants was associated with high costs and required long-term follow-ups. Our primary finding was that the highest costs occurred among infants born at lower GAs and that this relationship was difficult to detect unless an appropriate control group was used. Moreover, we identified a shortage of studies that estimated, in particular, costs that occurred outside hospitals and after the first year of life. Further research on BPD is required to understand the long-term consequences regarding healthcare costs and the social aspects of BPD.

**Acknowledgements** This systematic review was part of the Less is More project (Swedish Research Council grant number #2018–00770), a longitudinal study of clinical and economic outcomes in children born at very low gestational age. This study thus contributes with an understanding of the current knowledge and important gaps to the Less is More study design for long-term follow-up and economic evaluation.

**Contributors** All authors contributed to the design of the study. JH, CL and HG designed the database search and data extraction methods. JH and HG undertook the literature search, assessed studies for eligibility and extracted data. In case of disagreement, assessments were made in discussion with CL. DL and AH contributed clinical expertise on preterm infants and morbidity. JH, CL and HG discussed the data and interpreted the results. JH and HG drafted the manuscript. All authors critically reviewed and approved the final manuscript.

**Funding** JH was financed by the University of Gothenburg Centre for Person-Centred Care (GPCC) teaching assistant programme and the Mary von Sydow Foundation. CL was financed by the University of Gothenburg Centre for Person-Centred Care (GPCC). DL was financed by the Swedish Research Council (#2018–00770). AH was supported by the Swedish Research Council (#2016–01131), the Gothenburg Medical Society, De Blindas Vänner and Government grants under the ALF agreements (ALFGBG-717971 and ALFGBG-812951), and The Wallenberg Clinical Scholars. HG was financed by the Swedish Research Council (#2016–01131) and the Swedish Research Council (#2018–00770).

**Disclaimer** The funders played no role in the design of the study or writing the protocol.

**Competing interests** JH report no competing interests. CL holds stocks in Premalux AB. DL and AH hold stock/stock options in Premalux AB, and have received consulting fees from Takeda Inc. HG is employed part-time by Statfinn and EPID Research (part of IQVIA), which is a contract research organisation that performs commissioned pharmacoepidemiological studies, and thus its employees have been and currently are working in collaboration with several pharmaceutical companies.

**Patient consent for publication** Not required.

**Provenance and peer review** Not commissioned; externally peer reviewed.

**Data availability statement** All data relevant to the study are included in the article. Original data are available in the reviewed publications which are all cited. Additional data from the data extraction performed are available on reasonable request from the corresponding author.

**ORCID iDs**
Jhangir Humayun http://orcid.org/0000-0002-0507-8216
Ann Hellström http://orcid.org/0000-0002-9259-1244
Hanna Gyllensten http://orcid.org/0000-0001-6890-5162

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
