## [Reviewer comments · BMJ Open]

ARTICLE DETAILS

TITLE (PROVISIONAL)	SYSTEMATIC REVIEW OF THE HEALTHCARE COST OF BRONCHOPULMONARY DYSPLASIA
AUTHORS	Humayun, Jhangir; Löfqvist, Chatarina; Ley, David; Hellström, Ann; Gyllensten, Hanna

VERSION 1 – REVIEW

REVIEWER	Peter Auguste University of Warwick Warwick Medical School
REVIEW RETURNED	30-Nov-2020

GENERAL COMMENTS	Authors conducted a systematic review with the aim of identifying studies that reported the costs associated with treating bronchopulmonary dysplasia in babies. The authors included 13 studies in their review. There was considerable heterogeneity between the studies.  • The introduction is well written, with lots of clinical content and less so of economic content. Would it be helpful to include what therapies are available for the treatment of bronchopulmonary dysplasia in preterm babies? Also, the statement on page 5 lines 12-14 ‘An economic evaluation of healthcare costs is important for aiding decision-making bodies to prioritize health care funding effectively...’ needs rephrasing. Though this statement is partially correct, as the method of economic evaluation is important to aid decision-making. However, economic evaluation considers both the costs and the benefits of alternative courses of action, and not an ‘economic evaluation of healthcare costs.’ • Under the sub-heading Search strategy, the authors included a sentence about data extraction, which may be better placed in the Data extraction paragraph. Should the sub-heading Data extraction be extended to Data extraction and quality assessment? Did the authors undertake a scoping search of the literature, as the review did not identify any economic evaluation studies, so it might have not been appropriate to use the quality assessment tool by Evers and colleagues? • Caution should be taken when converting costs to one common currency and inflating to current prices. One example good example is shown here. The authors identified a study that was undertaken in 1981, inflating the costs to 2019 implicitly means that the management of BPD has not changed in 39 years. Is that a plausible assumption? • Please can the authors cite the studies as mentioned in the results section (page 6)? For example, in four studies, the definition for BPD included... Which four studies? • Little details are reported in table 1 with regards to costs. What intervention was used? What definition of bronchopulmonary dysplasia is being used in each study? What is the perspective and
--

	setting of each study included in the review? From Table 1, two studies used a model. What types of models were used in these studies? Are these costs per patient treated? What do the authors mean by time horizon? Time horizon in an economic evaluation context is the time duration of when the costs and benefits are estimated. In some cases, index hospitalization is reported under the time horizon sub-heading. Under the costs included, the authors state 'direct hospital costs', but this doesn't give any indication about what hospital costs are included. Furthermore, given that the review is about the costs of bronchopulmonary dysplasia, there should be some indication/mention of the resource use to derive the costs in each study. This may help to explain any similarities/differences seen in the costs reported.  • Given the methodological issues with meta-analysing cost/economic evaluation studies, results are generally discussed narratively. However, the manuscript does not provide enough content about each study or comparative analysis between the studies in the results section. Thanks for alluding to the limitations in the manuscript. However, it appears that these are related to both limitations of the review but more so the limitations of the studies. Would the authors consider that the type of quality appraisal tool used was a limitation of the study as well? Also, I would not consider the assessment of eligibility as a strength of the manuscript as this is part of the process of undertaking a systematic review. • The authors stated that further research on BPD is required. This statement is vague, so can the authors provide more information about the type of studies? Should these studies be retrospective or prospective? Can this information be derived from a database, for example?
--	---

REVIEWER	Kednapa Thavorn Institute for Clinical Evaluative Sciences, ICES @uOttawa
REVIEW RETURNED	29-Dec-2020

GENERAL COMMENTS	The authors conducted a systematic review of studies that reported the costs of bronchopulmonary dysplasia (BPD). The study identified 13 studies; most studies showed BPD's cost during index hospitalization and/or during the first year of life. The article reads well. I have a few clarification questions. Major comments  1. Could the authors please justify why only two electronic databases were used for this review? 2. This study is claimed to be a systematic review, but why were grey literature and references of the included studies not searched? 3. Table 1 should include a column indicating how each study defined BPD. 4. Table 1, why BPD costs reported in Greenough et al. in 2006 and 2010 were relatively low compared to other studies? 5. The study assessed the risk of bias of the included studies. Could the authors please elaborate on how the risk of bias may contribute to interpreting the results of this systematic review? 6. In addition to the costs reported in the included studies, it is essential to report and/or discuss the methods used to estimate BPD costs. Different costing methods can influence the cost estimates (Health Economics 18(4):377-88). Minor comments  1. An abstract should be revised. Further details should be added for the method and result sections. The result section should include the
--

	range of BPD costs and results of the risk of bias assessment. 2. On-Page 5, the authors indicated that "An economic evaluation of healthcare costs is important for aiding decision-making bodies...". Did the authors mean to say, "An economic evaluation of healthcare interventions"? There is no such thing as an economic evaluation of "healthcare costs."
--	--

VERSION 1 – AUTHOR RESPONSE

Reviewer: 1

Mr. Peter Auguste, University of Warwick Warwick Medical School Comments to the Author:

Authors conducted a systematic review with the aim of identifying studies that reported the costs associated with treating bronchopulmonary dysplasia in babies. The authors included 13 studies in their review. There was considerable heterogeneity between the studies.

- The introduction is well written, with lots of clinical content and less so of economic content. Would it be helpful to include what therapies are available for the treatment of bronchopulmonary dysplasia in preterm babies? Also, the statement on page 5 lines 12-14 ‘An economic evaluation of healthcare costs is important for aiding decision-making bodies to prioritize health care funding effectively...’ needs rephrasing. Though this statement is partially correct, as the method of economic evaluation is important to aid decision-making. However, economic evaluation considers both the costs and the benefits of alternative courses of action, and not an ‘economic evaluation of healthcare costs.’

Response: Thank you for your comments and feedback.

We added a sentence indicating the therapies available in the background, and a sentence at the end of the discussion indicating ongoing research on therapies for extremely preterm infants.

The erroneous sentence has been revised and now reads “An economic evaluation is important for aiding decision-making bodies to prioritize healthcare funding effectively and understand the associated long-term costs.”

- Under the sub-heading Search strategy, the authors included a sentence about data extraction, which may be better placed in the Data extraction paragraph. Should the sub-heading Data extraction be extended to Data extraction and quality assessment? Did the authors undertake a scoping search of the literature, as the review did not identify any economic evaluation studies, so it might have not been appropriate to use the quality assessment tool by Evers and colleagues?

Response: We have now moved the indicated sentence to the section about data extraction and changed the sub-heading to Data extraction and quality assessment, as suggested.

We agree that the choice of quality assessment tool is difficult. To our knowledge there is no consensus as to which tool should be used for the type of descriptive cost studies included in this review, and not even a consensus on the tool to use in economic evaluations (as is also clear from a recent publication in Value in Health, by Watts and Li, 2019. *Use of Checklists in Reviews of Health Economic Evaluations, 2010-2018*). We did find the chosen tool useful for assessing the studies and identify potential problems in how these were conducted, even if some of the items were of less relevance. Moreover, it is not that we actively excluded economic evaluations, but the studies identified during the search did not provide any cost data possible to include in this review, but only reported eg mean difference between groups

(without providing costs per group) or were excluded for other reasons (not reporting information based on BPD for example, but for a larger group of infants with many different diagnoses). Although the CHEERS checklist is becoming requested by more journals, it is clear that it will take some time before the main body of health economic literature actually addresses all the information asked for in that checklist, such as quantities and prices for resource use in each intervention group.

Regarding the type of review, our intention was not to conduct a scoping review and our focus was in the outcomes, ie the costs related to BPD, rather than on providing a general overview and mapping of all types of research to some extent associated with costs among infants with BPD. We believe that the issue under study here is already defined and described beforehand. Although there are few systematic scoping reviews in the field of health economics so far, we would argue that such a study would have a research question more in line with how are costs measured, rather than what is the actual cost and its distribution in different groups of infants.

- Caution should be taken when converting costs to one common currency and inflating to current prices. One example good example is shown here. The authors identified a study that was undertaken in 1981, inflating the costs to 2019 implicitly means that the management of BPD has not changed in 39 years. Is that a plausible assumption?

Response: We absolutely agree that much has happened since the early 80s, both in healthcare and in society, and thus using purchasing power parities to estimate the inflated costs should be viewed more as an indication or approximation of a current price level for the care provided at that time in history than an actual cost in today's health system. However, not inflating costs to a common value year would make comparisons over time less relevant. In many reviews these questions are easily solved by basing a choice of a more limited study period on some medical breakthrough that changes the care provided to the relevant patient population. In this study such a breakthrough exists, the development and use of surfactants, but that happened before the first identified studies of associated costs. However, the effect of the surfactant introduction on costs among all very-low-birth-weight infants (regardless BPD status) has previously been described by Schwartz et al, in the New England Journal of Medicine in 1994. We have now added a segment clarifying the problems associated with calculation and inflation of costs over long time periods in the fifth paragraph of the discussion, where we write about the variation in cost estimates across countries and over time.

- Please can the authors cite the studies as mentioned in the results section (page 6)? For example, in four studies, the definition for BPD included... Which four studies?

Response: The BPD definitions have been added to table 1, for added clarity, and the text now refers to that table.

- Little details are reported in table 1 with regards to costs. What intervention was used? What definition of bronchopulmonary dysplasia is being used in each study? What is the perspective and setting of each study included in the review? From Table 1, two studies used a model. What types of models were used in these studies? Are these costs per patient treated? What do the authors mean by time horizon? Time horizon in an economic evaluation context is the time duration of when the costs and benefits are estimated. In some cases, index hospitalization is reported under the time horizon sub-heading. Under the costs included, the authors state 'direct hospital costs', but this doesn't give any indication about what hospital costs are included. Furthermore, given that the review is about the costs of bronchopulmonary dysplasia, there should be some indication/mention of the resource use

to derive the costs in each study. This may help to explain any similarities/differences seen in the costs reported.

Response: Thanks for drawing our attention to the fact that we had oversimplified this table. We have now added information about definition of BPD, the models used, the denominator of the cost column (per child), clarified time horizon and age of children involved in each study, cost perspective and which costs were included as well as how. None of the studies include a formal experiment/intervention, although one compare outcomes prospectively between infants receiving own mothers milk v. those getting formula, and we added that information to the table.

- Given the methodological issues with meta-analysing cost/economic evaluation studies, results are generally discussed narratively. However, the manuscript does not provide enough content about each study or comparative analysis between the studies in the results section. Thanks for alluding to the limitations in the manuscript. However, it appears that these are related to both limitations of the review but more so the limitations of the studies. Would the authors consider that the type of quality appraisal tool used was a limitation of the study as well? Also, I would not consider the assessment of eligibility as a strength of the manuscript as this is part of the process of undertaking a systematic review.

Response: In addition to the described changes to table 1, we also revised the results section to provide a better understanding of what was included in each calculated cost. We also adjusted figures 2 and 3 to incorporate relevant changes.

We have revised the discussion based on your comment about limitations of the review vs limitations in included studies.

We believe the CHEC tool was useful for identifying limitations in the included studies, and have now elaborated a bit more in the section in the Results, under the heading Risk of bias in included studies, indicating which CHEC items were assessed to be fulfilled, and our overall assessment of bias in the included studies. It is now also referred to in the discussion.

We agree that conducting the study as planned is not a methodological strength, and that was not how the sentence was intended. We meant strength of evidence in comparison to previous reviews rather than methodological strength vs limitations and the sentence is now removed. Thanks for drawing our attention to how it got interpreted.

- The authors stated that further research on BPD is required. This statement is vague, so can the authors provide more information about the type of studies? Should these studies be retrospective or prospective? Can this information be derived from a database, for example?

Response: It has been clarified in the paragraph (second from last in the discussion) about future research that those studies called for, on long-term economic impact and costs outside of the health system, can probably to some extent be retrospective register studies, but there are some types of resources not available from administrative sources that would need to be collected from identified individuals/parents of affected infants and children, thus using prospective data collection methods.

Reviewer: 2

Dr. Kednapa Thavorn, Institute for Clinical Evaluative Sciences, The Ottawa Hospital
Research Institute Comments to the Author:

The authors conducted a systematic review of studies that reported the costs of bronchopulmonary dysplasia (BPD). The study identified 13 studies; most studies showed BPD's cost during index hospitalization and/or during the first year of life.

The article reads well. I have a few clarification questions.

Major comments

1. Could the authors please justify why only two electronic databases were used for this review?

Response: Thank you for your comments and feedback.

With regards to databases, these databases have previously been identified to cover well the evidence available from economic evaluations (unfortunately the HEED database ceased to be available a couple of years ago, see also Pitt et al 2015. *Economic Evaluation in Global Perspective: A Bibliometric Analysis of the Recent Literature*. In Health Economics.), and are often used in similar reviews (see eg Dangouloff et al 2020. *Systematic literature review of the economic burden of spinal muscular atrophy and economic evaluations of treatments*. In Orphanet J Rare Dis.). The search was developed by us researchers in collaboration with experienced librarians in our university library, but there is of course always a possibility that some studies were not identified through this strategy.

2. This study is claimed to be a systematic review, but why were grey literature and references of the included studies not searched?

Response: We have now conducted hand search of reference and citation of include studies and clarified that this was done in the manuscript. However, it did not identify any additional included studies.

However, there is a newer publication by Mowitz et al 2020 (covering a cohort of 1715 infants with BPD born 2009-2016), which was published online in December 2020, after our submission to the journal. In the study is extracted information from a smaller database than what is included in the included study by Mowitz et al 2019 (covering 4904 infants with BPD born 2009-2015). Both studies were sponsored by Shire/Takeda and the cost is similar (although not exactly the same due to the first study only reporting index hospitalization costs and the new study reporting costs during the first year or life). From an initial assessment that study, however, provides an estimate of the payments rather than costs or charges.

3. Table 1 should include a column indicating how each study defined BPD.

Response: The BPD definitions have been added to table 1, for added clarity, and the text now refers to that table.

4. Table 1, why BPD costs reported in Greenough et al. in 2006 and 2010 were relatively low compared to other studies?

Response: We have clarified in the results section about "Follow-up costs and home care" what differs between studies measuring costs during specific ages and those looking at costs from birth and onwards.

5. The study assessed the risk of bias of the included studies. Could the authors please elaborate on how the risk of bias may contribute to interpreting the results of this systematic review?

Response: Thank you for bringing to our attention that this was unclear. We now added a section in the Results, under the heading Risk of bias in included studies, indicating which

CHEC items were assessed to be fulfilled, and our overall assessment of bias in the included studies. It is now also referred to in the discussion.

6. In addition to the costs reported in the included studies, it is essential to report and/or discuss the methods used to estimate BPD costs. Different costing methods can influence the cost estimates (Health Economics 18(4):377-88).

Response: We have revised table 1. Thanks for bringing to our attention that we had oversimplified the table.

Minor comments

1. An abstract should be revised. Further details should be added for the method and result sections. The result section should include the range of BPD costs and results of the risk of bias assessment.

Response: The abstract has been revised to better describe the methods and results, as suggested.

2. On-Page 5, the authors indicated that "An economic evaluation of healthcare costs is important for aiding decision-making bodies...". Did the authors mean to say, "An economic evaluation of healthcare interventions"? There is no such thing as an economic evaluation of "healthcare costs."

Response: The erroneous sentence has been revised and now reads "An economic evaluation is important for aiding decision-making bodies to prioritize healthcare funding effectively and understand the associated long-term costs."

VERSION 2 – REVIEW

REVIEWER	Peter Auguste University of Warwick Warwick Medical School
REVIEW RETURNED	11-Mar-2021

GENERAL COMMENTS	The authors have submitted a revised manuscript by considering/addressing some of the concerns raised by the reviewers. However, there were still some minor concerns, which are detailed below. At times the authors have stated/mentioned that X studies reported Y but have not provided the references to these studies, so it leaves the reader uncertain which studies the authors are referring to. For example, on page 37 'only one study conducted a sensitivity analysis of their results.' Also, only two papers had more than 3 items missing...' but which studies? This is seen throughout the manuscript. Furthermore, page 37 lines 9-10 that sentence should be rephrased. Manuscript could benefit from further editing. Page 32 line 25, 'Costs were translated to USD,' Costs were converted to USD... Surprised at the paucity of economic evaluation studies in this area. Please can the authors justify why their study was not included in the systematic review?
---

VERSION 2 – AUTHOR RESPONSE

Reviewer Comments for the Author/revision notes:

Mr. Peter Auguste, University of Warwick Warwick Medical School Comments to the Author:

The authors have submitted a revised manuscript by considering/addressing some of the concerns raised by the reviewers. However, there were still some minor concerns, which are detailed below.

Response: Thank you for your comments and feedback.

At times the authors have stated/mentioned that X studies reported Y but have not provided the references to these studies, so it leaves the reader uncertain which studies the authors are referring to. For example, on page 37 'only one study conducted a sensitivity analysis of their results.' Also, only two papers had more than 3 items missing...' but which studies? This is seen throughout the manuscript.

Response: We initially found it a bit onerous to have all these references in the text and thus opted for keeping it in the tables, but agree that it will be easier to find the information when available in both text and table. References are now added to all such statements. AAWe also moved these references in the sentence so they occur in connection to where we refer to the number of studies. . Unfortunately, these changes in references werenot possible to track in a good way in the text. On a few occasions, we also clarified/corrected the number of studies now when adding the references.

Furthermore, page 37 lines 9-10 that sentence should be rephrased.

Response: The sentence "Of the 13 included studies, the checklist items judged relevant for each study method were to a large extent also covered in the paper." has been rephrased to "Checklist items judged relevant were largely covered by the included papers."

Manuscript could benefit from further editing. Page 32 line 25, 'Costs were translated to USD,' Costs were converted to USD...

Response: The sentence has been adjusted according to the suggestion, and the text has been revised throughout to improve clarity and ease of reading.

Surprised at the paucity of economic evaluation studies in this area. Please can the authors justify why their study was not included in the systematic review?

Response: We have an ongoing study on this topic, but it only just started recruiting last year, and thus results will be available in a couple of years, with cost data based on registers being available later than the health outcomes collected from e.g., infants medical records. So the main reason we initiated this systematic review was to make sure we had a full picture of the available knowledge. ,We already had an idea that there were fairly few studies available from our preliminary searches, but we were also a bit surprised by how little we could find.